# Unveiling the Genetic Architecture of Semen Traits in Thai Native Roosters: A Comprehensive Analysis Using Random Regression and Spline Function Models

**DOI:** 10.3390/ani14192853

**Published:** 2024-10-03

**Authors:** Iin Mulyawati Daryatmo, Jiraporn Juiputta, Vibuntita Chankitisakul, Wuttigrai Boonkum

**Affiliations:** 1Department of Animal Science, Faculty of Agriculture, Khon Kean University, Khon Kean 40002, Thailand; iin.d@kkumail.com (I.M.D.); jiraporn.ju@kkumail.com (J.J.); vibuch@kku.ac.th (V.C.); 2Network Center for Animal Breeding and Omics Research, Khon Kaen University, Khon Kaen 40002, Thailand

**Keywords:** spline function, heritability, genetic correlation, genetic parameter, indigenous chicken

## Abstract

**Simple Summary:**

Improving the genetic traits influencing rooster semen quality and quantity is crucial for optimizing poultry production. Breeders can identify heritable traits that significantly influence semen characteristics by evaluating genetic parameters, ensuring that only the best roosters are selected for breeding purposes. The development of appropriate genetic models aids in accurately predicting these traits, thereby enhancing selection efficiency. Selection based on these models boosts fertility and hatchability and promotes the propagation of desirable traits across generations. This approach leads to healthier, more productive flocks, increased economic efficiency, and more sustainable poultry farming practices.

**Abstract:**

Improving reproductive traits, particularly semen quality and quantity, is crucial for optimizing poultry production and addressing the current limitations in native chicken reproduction. The aim of this study was to develop a genetic model to estimate genetic parameters guiding the selection of individual Thai native roosters. Using data collected from 3475 records of 242 Thai native grandparent roosters aged 1–4 years, we evaluated semen traits (mass movement, semen volume, and sperm concentration) over 54 weeks. A random regression test–day model incorporating five covariance functions, including a linear spline function with four, five, six, and eight knots (SP4, SP5, SP6, and SP8) and second-order Legendre polynomial function (LG2), was used to estimate genetic parameters. The results showed that the SP8 model consistently outperformed the other models across all traits, with the lowest mean square error, highest coefficient of determination, and superior predictive ability. Heritability estimates for mass movement, semen volume, and sperm concentration ranged from 0.10 to 0.25, 0.22 to 0.25, and 0.11 to 0.24, respectively, indicating moderate genetic influence on these traits. Genetic correlations between semen volume and sperm concentration were highest in the SP8 model, highlighting a strong genetic association between these traits. The SP8 model also revealed a high genetic correlation between mass movement and semen volume, supporting the potential for selecting mass movement as a predictor of semen volume. In conclusion, this study highlights the effectiveness of random regression models with linear spline functions to evaluate the genetic parameters of semen traits in native Thai roosters. The SP8 model is a robust tool for breeders to enhance the reproductive performance of native Thai chickens, contributing to sustainable poultry production systems.

## 1. Introduction

Poultry in Thailand have significant genetic diversity, particularly among various local chicken breeds such as the Pradu Hang Dum, which is renowned for its strong disease resistance, adaptability to environmental conditions, and high-quality meat [1,2,3]. Native chickens are vital to rural farming systems and play a crucial role in supporting the economy of rural households by providing an additional source of income and easily accessible animal protein [4]. These native breeds are an important genetic resource for global food security due to their adaptability and diversity [5,6]. In recent decades, significant progress has been made in improving the production traits of native chickens, particularly growth performance and egg production [7,8,9]. However, reproductive traits are often overlooked, resulting in reduced fertility and lower chick production. Semen traits are among the most important factors affecting chicken reproductive performance; however, they frequently receive insufficient attention during the genetic development of native chickens [10,11]. Addressing this issue requires significant improvements in semen quantity and quality, which would directly contribute to increasing bird populations and enhance feed production to meet the needs of the growing native chicken population. In this context, more in-depth research and targeted genetic development strategies are essential to improve overall reproductive performance. This is critical for the sustainability of native chicken production systems in Thailand and other regions where these strategies can be applied.

Reproductive traits, including semen quality and quantity, are crucial for the sustainability and efficiency of native chicken populations [2,12,13]. Neglecting semen traits can hinder production efforts by leading to low fertilization rates and reduced artificial insemination success [14,15]. Research by Mohan et al. [16] and Tarif et al. [17] indicates that artificial insemination is the primary method for boosting native chicken production and a key tool for genetic improvement. However, achieving this also depends significantly on the male’s ability to produce high-quality semen. Semen with sufficient volume, strong sperm mass movement, and high sperm concentration are essential to achieve optimal fertilization rates [18,19]. Genetic factors significantly influence semen traits, and the genetic superiority of males is a critical determinant of reproductive success [20,21,22]. In recent decades, the reproductive performance of roosters has declined primarily because of a lack of focus on semen quality and quantity [23,24]. Therefore, enhancing semen traits through genetic approaches is essential to support effective breeding programs and ensure the sustainability and long-term reproductive health of native chickens.

Estimating genetic parameters is a highly effective and widely utilized method in livestock genetics [25,26,27]. This approach is vital for livestock breeding programs because it enables farmers and researchers to understand how traits such as growth, meat production, and egg yield are inherited across generations [28]. Breeders can make informed decisions regarding which animals to breed to achieve optimal results by estimating parameters such as heritability and genetic correlations. Additionally, this method is extensively used because it facilitates the prediction of future performance, thereby enhancing the overall quality and productivity of native chicken populations. This is crucial for improving the efficiency and effectiveness of breeding programs. The random regression model (RRM) is the most popular tool used to estimate genetic parameters in animals. RRM offers a valuable approach to semen quality evaluation by allowing for dynamic analysis of longitudinal data on semen changes over time and under varying environmental conditions, as demonstrated in several studies [28,29,30]. This model effectively separates genetic effects from environmental influences, thus supporting more efficient genetic selection [31]. The use of RRM to evaluate the genetics of broiler and layer chickens is becoming increasingly widespread across various countries as research advances [32,33]. In contrast, applying genetic models to native chickens is less common, with fewer published studies than those on broiler and layer chickens. This is primarily because native chickens are typically raised in small, diverse farming systems where breeding goals and management practices vary widely. Unlike commercial broiler and layer chickens, which have well-established breeding programs and uniform management, native chickens are generally not the focus of large-scale genetic improvements. Moreover, economic incentives for intensive research and development of native chicken breeds are low, resulting in less investment in this area.

Considering the evolving trends in native chicken production and the growing number of farms in developing countries, such as Thailand, it is crucial to explore the potential of RRM for evaluating the genetic traits of native chicken semen. This method can improve the identification of males with superior genetics and support more effective selection programs, ultimately enhancing livestock productivity and sustainability. Therefore, the objective of this study was to identify a suitable genetic model for estimating genetic parameters for the genetic selection of individual Thai native chickens.

## 2. Materials and Methods

### 2.1. Animal Ethics and Animal Management

This study was reviewed and approved by the Institutional Animal Care and Use Committee of Khon Kaen University in accordance with the Ethics of Animal Experimentation guidelines set by the National Research Council of Thailand (Approval No. IACUC-KKU-114/66; 6 October 2023). This study was conducted at the experimental farm of the Network Center for Animal Breeding and Omics Research, Faculty of Agriculture, Khon Kaen University, Thailand. The data consisted of 3475 records of 242 Thai native grandparent roosters (Pradu Hang Dum), aged 1–4 years, housed in 45 × 50 × 60 cm cages within an open house system and exposed to natural sunlight and ambient temperature. Each rooster was provided with approximately 110 g of commercial breeder feed per day (containing 90.07% dry matter, 17.15% crude protein, 3.35% crude fiber, 3.99% ether extract, and 9.75% ash), with ad libitum access to drinking water throughout the experimental period. Additional recorded data included rooster identification (ID), body weight, age, ambient temperature and relative humidity, month and year of birth, and semen data collection.

### 2.2. Semen Collection and Evaluation

Semen samples were collected every Saturday for 54 weeks using the dorsal abdominal massage technique [34]. Each sample was carefully transferred into a 1.5-mL Eppendorf tube containing 0.1 mL of IGGKPh diluent [35]. The semen samples were protected from light and maintained at a temperature of 22–25 °C during transport, which was completed within 20 min of collection, to ensure proper handling. Standard semen analysis procedures, including macro- and microscopic evaluations, were conducted. The same person consistently collected the semen to ensure optimal quality and quantity, and meticulous care was taken to avoid cross-contamination. Semen characteristics such as mass movement, semen volume, and sperm concentration were assessed. The volume was measured using a graded 1 mL syringe. Sperm motility was evaluated based on mass movement. A drop of semen was placed on a slide without a coverslip, observed under a compound microscope at 100× magnification, and scored on a 1–5-point scale following the method described by Peters et al. [36]. Finally, sperm concentration was determined using a hemocytometer. A 1 µL semen sample was diluted with 999 µL of 4% sodium chloride, and a drop of this diluted sample was placed on a hemocytometer. The sperm concentration in 1 mL of semen was calculated via observation under a compound microscope at 400× magnification.

### 2.3. Genetic Model and Statistical Analysis

Data collected from the experimental farms at Khon Kaen University were validated before genetic analysis using the Proc UNIVARIATE procedure in SAS v.9.0 software. This validation step ensured proper data distribution by checking for normality, homogeneity of variance, and identifying outliers (values outside ± 3 standard deviations). Variance components and genetic parameters, including heritability, repeatability, genetic correlations, and phenotypic correlations, were estimated using a multiple-trait random regression test–day model. This model incorporated five covariance functions, i.e., four linear spline functions (SP4, SP5, SP6, and SP8 knots) and a second-order Legendre polynomial function (LG2, used as a control covariance function based on the results of Daryatmo et al. [37]). These were analyzed using the average information expectation maximization restricted maximum likelihood (AI-REML) approach with the AI-REMLF90 program [38]. The model used for the analysis can be defined as follows:$$y_{ijklm}={HMY}_{i}+{AGE}_{j}+{BW}_{k}+\sum_{m=0}^{q}a_{lm}Z_{m\left(t\right)}+\sum_{m=0}^{q}p_{lm}Z_{m\left(t\right)}+e_{ijklm}$$
where $y_{ijklm}$ represents the observation value of test–day semen traits at each time point, ${HMY}_{i}$ denotes the fixed effect of the combination of the chicken hatch set and test month and year of data collection, ${AGE}_{j}$ represents the fixed effect of rooster age, ${BW}_{k}$ indicates the fixed effect of rooster body weight, $a_{lm}$ is the random regression coefficient for the additive genetic effects of rooster l, and $p_{lm}$ is the random regression coefficient for the permanent environmental effects of rooster l, and $e_{ijklm}$ is the random residual effect for each observation. $Z_{m(t)}$ represents the value of the coefficients of the covariance functions at test–day semen collection period t, and q is the number of coefficients of the covariance functions. Covariance functions were equally designed for additive genetic and permanent environmental effects, depending on the number of knots and orders. The covariance matrix for all models was as follows:$$\mathrm{Var}\left[\begin{array}{c} a\\ p\\ e \end{array}\right]=\left[\begin{array}{ccc} G⨂A & 0 & 0\\ 0 & P⨂I & 0\\ 0 & 0 & R \end{array}\right]$$
where G and P are the covariance matrices for additive genetic and permanent environmental effects, respectively, A is the additive genetic relationship matrix among animals, I is an identity matrix, ⨂ is the Kronecker product between matrices, and R is the diagonal matrix of residual variances allowed to differ for test–day semen collection. The covariance functions used in the analysis were as follows:

Linear spline functions at 4, 5, 6, and 8 knots (SP4, SP5, SP6, and SP8, respectively) were as follows (Misztal, [39]):SP4: $f\left(t\right)=Z_{1}\left(t_{9}\right)+Z_{2}\left(t_{25}\right)+Z_{3}\left(t_{37}\right)+Z_{4}\left(t_{54}\right)$SP5: $f\left(t\right)=Z_{1}(t_{9})+Z_{2}(t_{17})+Z_{3}(t_{25})+Z_{4}(t_{37})+Z_{5}(t_{54})$SP6: $f\left(t\right)=Z_{1}\left(t_{9}\right)+Z_{2}\left(t_{17}\right)+Z_{3}\left(t_{25}\right)+Z_{4}\left(t_{37}\right)+Z_{5}\left(t_{45}\right)+Z_{6}(t_{54})$SP8: $f\left(t\right)=Z_{1}\left(t_{9}\right)+Z_{2}\left(t_{17}\right)+Z_{3}\left(t_{25}\right)+{Z_{4}\left(t_{29}\right)+Z}_{5}\left(t_{37}\right)+Z_{6}\left(t_{41}\right)+Z_{7}\left(t_{45}\right)+Z_{8}(t_{54})$For $T_{i}\leq t<T_{i+1}$: $Z_{i}\left(t\right)=\frac{t-T_{i}}{T_{i+1}-T_{i}},\;Z_{i+1}\left(t\right)=\frac{T_{i+1}-t}{T_{i+1}-T_{i}}$, and $Z_{j}=0,\;where\;j<i\;and\;j>i+1$; For $T_{i}=t:Z_{i}\left(t\right)=0\;and\;Z_{j}=0\;where\;j\neq i$,where $T_{i}$ is rooster age (months) in semen collection at the knot i^th^; t is the semen collection according to rooster age (months), situated between knots $T_{i+1}$ and $T_{i}$.For semen collection, the placement of knots was based on phenotypic characteristics observed during specific months. Various knot positions were identified and labeled SP4 to SP8, corresponding to the stages of deteriorating semen quality. For SP4, knots were placed at a roosted age of 9, 25, 37, and 54 months; SP5 knots were placed at 9, 17, 25, 37, and 54 months; SP6 knots were placed at 9, 17, 25, 37, 45, and 54 months; and SP8 knots were positioned at 9, 17, 25, 29, 37, 41, 45, and 54 months.The second-order Legendre polynomial function (LG2; Gengler et al. [40]) was as follows:LG2: $f\left(t\right)=L_{1}+L_{2}+L_{3}$where $L_{1}=1$, $L_{2}=\sqrt{3}L$, $L_{3}=\sqrt{\frac{5}{4}}(3L^{2}-1)$, $L=-1+2\frac{\left({t-t}_{min}\right)}{\left(t_{max}-t_{min}\right)}$, t is the current month of semen data collection, $t_{min}$ is the first month of semen data collection, and $t_{max}$ is the last month of semen data collection.

### 2.4. Genetic Model Selection Criteria

The RRMs with five covariance functions (i.e., LG2, SP4, SP5, SP6, and SP8) were compared to select the most appropriate genetic and best-fitted model to describe the genetic parameters, heritability curve throughout the experiment, and genetic and phenotypic correlations using three criteria. (1) Goodness of fit criteria using mean square error (MSE), the coefficient of determination (*R*^2^), twice the negative log-likelihood (−2logL), and Akaike’s information criterion (AIC) from the entire data set, in which the lowest MSE, −2logL, and AIC and highest *R*^2^ indicate the best-fit model. The MSE and *R*^2^ are defined as MSE = $\frac{SSE}{n-p}$, where SSE is the error sum of the square, n is the number of observations, p is the number of model parameters, *R*^2^ = $\frac{SSE}{SST}$, where SST is the total sum of squares, −2logL = $-2log(p\left(y|\hat{\theta }\right))$, where θ denotes the vector of the model parameters, $(p\left(y|\hat{\theta }\right)$ is the likelihood of the data y evaluated at the maximum likelihood estimate $\hat{\theta }$, and AIC = −2logL+2p, where p is the number of model parameters. (2) The predictive ability in terms of $\rho_{y_{i},\hat{y_{i}}}$ through cross-validation, in which higher values indicate better predictive ability. (3) The heritability value (h^2^), when using any genetic model that gives a high heritability value, means that most of the variation in the trait is due to genetic factors. This makes genetic selection more efficient because the traits selected are more likely to be passed on to the next generation and can also lead to a reduction in the generation interval (the average age of parents when their offspring are born) because it allows for early selection. In addition, high heritability allows breeders to predict the genetic gain resulting from selection more accurately.

## 3. Results

### 3.1. Descriptive Statistics of Semen Traits

The average monthly semen traits (mass movement, semen volume, and sperm concentration) and regression analysis according to rooster age are shown in Figure 1. Regression analysis revealed a decline in mass movement, semen volume, and sperm concentration in aged roosters. Specifically, mass movement decreased with a slope of −0.0469 per month and a coefficient of determination (*R*^2^) of 0.5774. Semen volume also declined with age, with a slope of −0.0022 and an R^2^ of 0.3860. Similarly, sperm concentration decreased over time, with a slope of −0.0329 and *R*^2^ of 0.3732. The mean values and standard deviations for these traits were 2.93 ± 1.18 for mass movement, 0.35 ± 0.19 mL for semen volume, and 3.12 ± 1.50 × 10^9^/mL for sperm concentration.

### 3.2. Selection of the Optimum Model and Heritability Values

The results of comparing the RRM with the five covariance functions for semen traits in native Thai roosters are presented in Table 1. Overall, SP8 consistently yielded the best results across all semen traits, as reflected by its lower MSE, higher *R*^2^, superior predictive ability, and higher heritability values, making it the most reliable model for evaluating semen traits in native Thai roosters. For the trait of mass movement, the SP8 model exhibited the lowest MSE of 9.346 and the highest coefficient of determination (*R*^2^) of 0.559. The predictive ability of this model was 0.842, which was the highest among all of the models, and it also had the highest heritability (h^2^) of 0.118. In addition, the −2logL and AIC values were the lowest, indicating a strong fit. In the case of semen volume, the SP8 model performed the best, with an MSE of 7.209 and an *R*^2^ of 0.347. The predictive ability was 0.844, which was the highest in this category, and heritability was 0.238. The −2logL and AIC values were the lowest at 0, demonstrating that this model was the most appropriate for semen volume prediction. For sperm concentration, the SP8 model showed superior performance, with an MSE of 9.430 and an *R*^2^ of 0.362. This model showed the highest predictive ability (0.834) and heritability (0.133). Therefore, the SP8 model was the best-fitting model in this study.

### 3.3. Heritability Estimates

Heritability estimates of the semen traits in Thai native chickens were determined using RRMs with five covariance functions (Figure 2). Heritability estimates for all semen traits (mass movement, semen volume, and sperm concentration) were plotted based on the age of the roosters (9–53 months old). The heritability estimates for mass movement, semen volume, and sperm concentration were relatively consistent across the different models and ranged from 0.10 to 0.25, 0.22 to 0.25, and 0.11 to 0.24, respectively. Different covariance functions provided slightly different heritability estimates. For instance, higher-knot spline functions (SP8) tended to produce more variable heritability estimates across different ages than lower-knot functions (SP4). The eight-knot linear spline (SP8) model generally showed slightly higher heritability estimates for sperm concentration than the other models, particularly in older roosters.

### 3.4. Genetic and Phenotypic Correlations

The results presented in Table 2 provided a comparative analysis of the genetic and phenotypic correlations among mass movement, semen volume, and sperm concentration using different covariance functions: LG2 (second-order Legendre polynomial function) and SP4, SP5, SP6, and SP8 (linear spline functions at 4, 5, 6, and 8 knots, respectively). Among the models, SP8 demonstrated the highest genetic correlation between semen volume and sperm concentration (0.755), followed by SP6 (0.745), indicating a stronger genetic association between these traits in the spline models than in the polynomial models (LG2). Phenotypic correlations showed a similar pattern, with SP8 showing the highest correlation between semen volume and sperm concentration (0.729). Genetic correlations between mass movement and semen volume were highest in the SP8 model (0.552) and lowest in the SP4 model (0.501), whereas phenotypic correlations were also highest for SP8 (0.590). For sperm concentration, the SP8 model showed the highest genetic correlation (0.644) and phenotypic correlation (0.689) with mass movement. Overall, the spline models, particularly SP8, outperformed the Legendre polynomial function (LG2) in capturing stronger genetic and phenotypic correlations among the evaluated traits.

## 4. Discussion

Semen traits are crucial in poultry breeding and reproduction. Understanding these traits can lead to improved breeding strategies, enhanced fertility, and better overall productivity of native Thai roosters. Estimating genetic parameters is vital for breeders to select desirable traits. This study provides insights into heritability and genetic correlations. The mention of a random regression test–day model and linear spline functions indicates the need for sophisticated statistical techniques. This can improve the accuracy of findings and allow for a more nuanced analysis of genetic data over time.

The semen traits of Thai native roosters, particularly mass movement, semen volume, and sperm concentration, are crucial for assessing their breeding potential and overall productivity. In this study (Figure 1), the average and standard deviation values were 2.93 ± 1.18, 0.35 ± 0.19 mL, and 3.12 ± 1.50 × 10^9^/mL for mass movement, semen volume, and sperm concentration traits, respectively. Our results on semen characteristics are in agreement with several reports, such as those on the Beijing-You chickens of China [22], Aseel and Rhode Island Red chicken breeds [10], and seven chicken breeds raised in Nigeria [36]. These results provide important insights into the reproductive efficiency of these birds. Mass transfer is a key indicator of sperm motility and viability. Previous studies have shown a positive correlation between mass movement and fertility rates in poultry [41,42]. The average mass movement observed in this study is consistent with findings in other breeds, suggesting that native Thai roosters have good sperm motility, which is essential for successful fertilization. Semen volume is another important factor affecting reproductive success. The average semen volume of 0.35 ± 0.19 mL reported in this study aligns with results from other chicken breeds [9,21,33]. Semen volume can be influenced by factors such as age, nutrition, and breeding practices [43,44,45]. Optimizing these factors can help enhance semen production, thereby improving overall fertility rates [41]. Sperm concentration is a critical determinant of fertility because higher sperm concentrations generally lead to a greater chance of successful fertilization [46]. The average sperm concentration of 3.12 ± 1.50 × 10⁹/mL found in Thai native roosters indicates strong reproductive potential. Our results are similar to and/or higher than those in other native chicken breeds, including three strains of native chickens from Bangladesh [47], in crossbred Korean native chickens [48], seven breeds of native chickens from Nigeria [36], and three Italian breeds [49]. Increasing sperm concentration through selective breeding and better management practices can further improve the reproductive efficiency of these roosters [46,50].

In this study, five covariance functions were compared using a random regression model to identify the most effective model for assessing semen traits. The results indicated that the SP8 model consistently outperformed the other models across all analyzed traits, including mass movement, semen volume, and sperm concentration. This finding aligns with previous research emphasizing the importance of selecting appropriate statistical models to accurately estimate heritability and predictive abilities in animal breeding [51,52,53,54]. The SP8 model exhibited the lowest MSE and highest coefficient of determination (*R*^2^) for mass movement, with values of 9.346 and 0.559, respectively. These results suggest that the SP8 model is a strong fit for predicting the mass movement of semen traits. A predictive ability of 0.842 further underscores the reliability of the model, as it indicates a high level of accuracy in predicting traits based on the data provided. The heritability estimate of 0.118 suggests that there is a moderate genetic component of this trait, which is consistent with the findings of other poultry studies. In terms of semen volume, the SP8 model again demonstrated superior performance, achieving an MSE of 7.209 and an *R*^2^ of 0.347. A predictive ability of 0.844 and a heritability estimate of 0.238 indicate that semen volume is influenced by both genetic and environmental factors, reinforcing the need for targeted breeding strategies. The low −2logL and AIC values further validated the SP8 model as the most appropriate choice for predicting semen volume. Previous studies have shown that the accurate estimation of semen volume is crucial for optimizing reproductive performance in poultry [16,55]. For sperm concentration, the SP8 model maintained its superiority, yielding an MSE of 9.430 and an *R*^2^ of 0.362. The predictive ability of 0.834 suggests that this model can effectively predict sperm concentrations based on the assessed traits. The heritability estimate of 0.133 indicates a lower genetic influence than mass movement and semen volume, suggesting a greater environmental impact on this trait. This observation is supported by King’ori et al. [56] and Sonseeda et al. [15], who noted that environmental factors often play a significant role in sperm quality and concentration in poultry. The consistent performance of the SP8 model across all semen traits highlight its robustness and reliability for evaluating these critical reproductive parameters. These findings suggest that the SP8 model is a valuable tool for breeders who aim to enhance the reproductive performance of native Thai roosters.

As breeding programs increasingly rely on genetic evaluations to make informed decisions, the ability to accurately predict semen traits is paramount [57]. The implications of these findings extend beyond the context of the present study. The successful application of the SP8 model in predicting semen traits could pave the way for improved breeding strategies, not only for Thai native roosters but also for other poultry species. Breeders can make more informed selections, ultimately leading to enhanced productivity and efficiency in poultry production systems, by utilizing models that accurately reflect the genetic architecture of traits [58]. In conclusion, the results of the present study underscore the importance of selecting appropriate statistical models to evaluate semen traits in native Thai roosters. The SP8 model emerged as the best-fit model, demonstrating superior predictive ability, a lower MSE, and higher heritability values across all traits assessed. These findings provide a solid foundation for future research to optimize breeding programs and enhance the reproductive performance of poultry. Continued exploration of genetic and environmental influences on semen traits is essential for advancing our understanding of avian reproduction and improving breeding outcomes.

The heritability of semen traits in poultry, especially native chickens (Table 1 and Figure 2), has recently gained attention because of its importance in breeding programs aimed at improving reproductive performance. The heritability estimates found in this study were similar to those previously reported by Thepnarong et al. [59] for the semen characteristics (semen volume, mass movement, sperm concentration, and total sperm) of Betong chickens (h^2^ = 0.04–0.12) and by Wolc et al. [60] for the sperm motility and sperm count of White Leghorn roosters (h^2^ = 0.08 and 0.13). However, the results of this study were lower than those of many previous studies, such as in Chinese male chickens, which focused on seven characteristics (semen volume, semen pH, semen color, sperm viability, sperm motility, sperm deformities, and sperm concentration) and showed that the heritability values ranged from 0.03 to 0.85 [22]. The estimated heritability values suggested that semen volume is a moderately heritable trait that can be effectively improved through selective breeding. Similarly, the heritability of sperm concentration has been reported in other studies, with estimates typically ranging from 0.10 to 0.46 [61,62]. These findings provide a genetic basis for these traits, enabling potential improvements through selective breeding. Additionally, the consistency of heritability estimates across different covariance models in this study supports the findings of other studies that emphasize the importance of choosing the right model for heritability estimation. For example, Plaengkaeo et al. [63], Mookprom et al. [52], and Eilers and Marx [64] showed that different covariance structures can produce varying heritability estimates, similar to the observation that higher-knot spline functions provide more variable estimates than lower-knot functions. This highlights the importance of carefully selecting the statistical models used in genetic evaluations. Although many of the heritability estimates in this study were consistent with those of previous studies, some results differed, particularly regarding age-related changes in heritability. The eight-knot linear spline (SP8) model showed slightly higher heritability estimates for sperm concentration in older roosters than the other models. This finding contrasts with that of Wolc et al. [60], which showed that heritability estimates for sperm traits tend to decrease with age. These differences might be due to underlying biological mechanisms such as genetic interactions and environmental influences that affect reproductive performance. These heritability estimates have important implications for breeding programs. The moderate heritability of semen traits suggests that selective breeding can lead to significant genetic improvements over time. For example, breeding programs that focus on increasing sperm concentration and semen volume could exploit the heritable nature of these traits to improve the reproductive efficiency of poultry production systems. However, the variability in heritability estimates based on age and model choice highlighted the need for further research to refine breeding strategies.

The comparative analysis of the genetic and phenotypic correlations presented in Table 2 highlights the efficacy of different covariance functions in elucidating the relationships between mass movement, semen volume, and sperm concentration. The results indicate that spline models, particularly the SP8 function, significantly outperform the Legendre polynomial function (LG2) in capturing these associations. This finding aligns with previous studies advocating the use of spline functions to model complex biological relationships more effectively [65]. Moreover, Mota et al. [66] revealed that using B-spline functions with four residual variance classes and segments was the best fit for the genetic evaluation of growth traits in meat-type quail, whereas Legendre polynomials underestimated the residual variance. In addition, Pereira et al. [67] reported that linear spline functions with six knots provided the lowest sum of residual variances across lactation. However, third-order Legendre polynomials were better suited for capturing additive genetic and permanent environmental effects.

The high genetic correlation observed between semen volume and sperm concentration (0.755) using the SP8 model suggests a strong underlying genetic basis for these traits. This is consistent with the literature, which emphasizes the importance of semen quality metrics in reproductive performance [22,68]. The robust genetic association indicates that improvements in one trait may lead to enhancements in the other, providing valuable insights for breeding programs aimed at optimizing reproductive efficiency. Furthermore, the phenotypic correlation of 0.729 between semen volume and sperm concentration reinforces the significance of these traits in practical applications. Higher semen volume is often associated with increased sperm concentration, which can enhance fertility outcomes. The findings of this study suggest that breeding strategies focusing on these correlated traits could yield substantial improvements in reproductive success. In examining the genetic correlations between mass movement and semen volume, the SP8 model again demonstrated the strongest association (0.552) when examining genetic correlations between mass movement and semen volume. This correlation highlights the potential for selecting mass movement as a predictor of semen volume, a relationship that has received limited attention in the literature. The importance of mass movement as a trait linked to fertility and overall sperm quality has been underscored by various researchers [69,70]. The current findings provide further evidence that mass movement could serve as a valuable criterion in breeding programs focused on enhancing reproductive traits. The phenotypic correlation of 0.590 between mass movement and semen volume is also noteworthy. It suggests that the environmental factors influencing these traits may be shared, warranting further investigation into the underlying mechanisms. Understanding how environmental conditions affect mass movement and semen volume could lead to improved management practices in breeding programs [15,71,72]. The highest genetic correlation for sperm concentration with mass movement (0.644) in the SP8 model also emphasizes the interconnectedness of these traits. The phenotypic correlation of 0.689 further supports the idea that sperm concentration is influenced by both genetic and environmental factors. This dual influence suggests that while genetic selection is crucial, environmental management practices must also be optimized to achieve the best outcomes in reproductive performance. Overall, the findings underscore the advantages of using spline models, particularly SP8, in analyzing genetic and phenotypic correlations among reproductive traits. These results advocate for integrating these models into breeding strategies to enhance the accuracy and efficiency of selection processes. Future research should focus on validating these correlations in larger populations and exploring the genetic mechanisms underlying these associations. In conclusion, the present study contributes to the understanding of the complex relationships between mass movement, semen volume, and sperm concentration. The superior performance of the spline models in capturing these correlations provides a robust framework for future research and breeding applications aimed at improving livestock reproductive traits. When using the SP8 model, several factors must be considered. The linear spline with eight knots is more complex than simpler models like SP4 or SP5, which can make it more computationally demanding and harder to implement with smaller datasets or limited computing resources. Additionally, the increased number of knots may lead to overfitting, particularly with datasets that have fewer observations, potentially limiting the model’s generalizability to other populations or time periods. Therefore, it is essential to validate the model’s goodness-of-fit to ensure its suitability for estimating genetic parameters in animal breeding.

Although the genetic models used in this study indicate that the estimated genetic parameters were still low, integrating other approaches, such as marker-assisted selection (MAS), quantitative trait loci (QTL), or genomic selection (GS), along with improving nutrition, health management, and environmental factors, may enhance the expression of genetic potential. By combining these methods, the effectiveness of breeding programs can be improved even when genetic parameters are low, ensuring continued progress in selecting superior animal breeds.

## 5. Conclusions

The results of this study confirm that spline functions provide the best fit for estimating the genetic parameters of semen traits in Thai native roosters. A random regression test–day model with an eight knots linear spline function best described the heritability curve over the semen collection period. Therefore, it is possible to obtain estimated breeding values to improve both the quality and quantity of semen in breeding programs for Thai native chickens. These findings will improve selection strategies for reproductive performance, ultimately benefiting the poultry industry and promoting sustainable practices. Moreover, the implications of this research extend beyond local applications, providing valuable insights for genetic studies in various poultry species worldwide.

## Figures and Tables

**Figure 1 animals-14-02853-f001:**
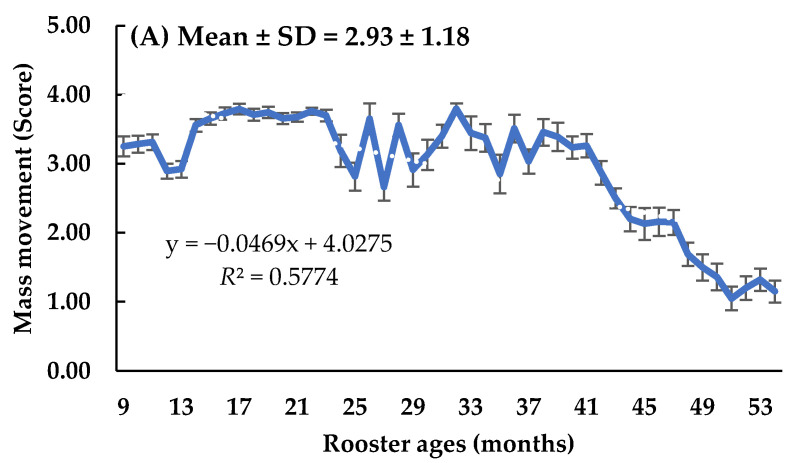
Average monthly semen traits (mass movement, semen volume, and sperm concentration) and linear regression analysis by rooster age.

**Figure 2 animals-14-02853-f002:**
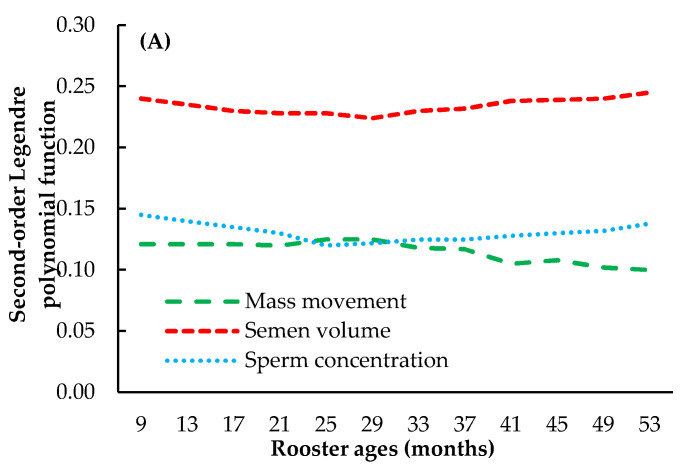
Estimated heritability of semen traits (mass movement, semen volume, and sperm concentration) using a random regression model with five covariance functions ((**A**) = second-order Legendre polynomial function (LG2); (**B**) = 4-knot linear spline function (SP4); (**C**) = 5-knot linear spline function (SP5); (**D**) = 6-knot linear spline function (SP6); (**E**) = 8-knot linear spline function (SP8)).

**Table 1 animals-14-02853-t001:** Comparison of statistics criteria of the random regression model with five covariance functions of semen traits in Thai native roosters.

Trait	Model	MSE	*R* ^2^	−2logL	AIC	$\rho_{y_{i},\hat{y_{i}}}$	h^2^
Mass movement	LG2	9.862	0.505	19	21	0.750	0.115
	SP4	10.357	0.499	25	27	0.742	0.114
	SP5	9.762	0.527	12	15	0.765	0.115
	SP6	9.350	0.555	8	9	0.825	0.117
	SP8	9.346	0.559	0	0	0.842	0.118
Semen volume	LG2	8.232	0.308	19	21	0.790	0.233
	SP4	8.526	0.288	25	27	0.782	0.230
	SP5	7.822	0.332	12	15	0.792	0.232
	SP6	7.217	0.343	8	9	0.826	0.236
	SP8	7.209	0.347	0	0	0.844	0.238
Sperm concentration	LG2	10.152	0.334	19	21	0.782	0.130
	SP4	11.029	0.325	25	27	0.777	0.121
	SP5	9.753	0.340	12	15	0.799	0.130
	SP6	9.450	0.355	8	9	0.815	0.133
	SP8	9.430	0.362	0	0	0.834	0.133

LG2 = second-order Legendre polynomial function; SP4, SP5, SP6, and SP8 = linear spline functions at 4, 5, 6, and 8 knots; MSE = mean square error; *R*^2^ = the coefficient of determination; −2logL = twice the negative log-likelihood; AIC = Akaike’s information criterion; $\rho_{y_{i},\hat{y_{i}}}$ = the predictive ability; h^2^ = heritability.

**Table 2 animals-14-02853-t002:** Genetic (above the diagonal) and phenotypic correlations (below the diagonal) between mass movement, semen volume, and sperm concentration using a random regression model with five covariance functions.

Model	Trait	Mass Movement	Semen Volume	Sperm Concentration
LG2	**Mass movement**	-	0.522	0.589
	**Semen volume**	0.565	-	0.629
	**Sperm concentration**	0.644	0.688	-
SP4	**Mass movement**	-	0.501	0.515
	**Semen volume**	0.515	-	0.546
	**Sperm concentration**	0.530	0.577	-
SP5	**Mass movement**	-	0.538	0.614
	**Semen volume**	0.577	-	0.672
	**Sperm concentration**	0.635	0.732	-
SP6	**Mass movement**	-	0.545	0.638
	**Semen volume**	0.580	-	0.699
	**Sperm concentration**	0.677	0.745	-
SP8	**Mass movement**	-	0.552	0.644
	**Semen volume**	0.590	-	0.729
	**Sperm concentration**	0.689	0.755	-

LG2 = second-order Legendre polynomial function; SP4, SP5, SP6, and SP8 = linear spline functions at 4, 5, 6, and 8 knots.

## Data Availability

Additional data are available from the corresponding authors upon request.

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
