# Peer review of "Unveiling the Genetic Architecture of Semen Traits in Thai Native Roosters: A Comprehensive Analysis Using Random Regression and Spline Function Models"

_animals, 2024, doi:10.3390/ani14192853_

Round 1

Reviewer 1 Report

Comments and Suggestions for Authors

Title:

Unveiling the Genetic Architecture of Semen Traits in Thai Native Chickens: A Comprehensive Analysis Using Random Regression and Spline Function Models

General Comments:

The article "Unveiling the Genetic Architecture of Semen Traits in Thai Native Chickens: A Comprehensive Analysis Using Random Regression and Spline Function Models" provides valuable insights into the semen traits of Thai native chickens using various statistical models. However, the following points need to be addressed:

·       How can the findings of this study be incorporated into breeding programs?

·       What are the limitations of the SP8 model compared to other models?

·       Are the findings of this study applicable to different poultry species and breeding environments?

Simple summary:

Lines 11-12: It should be written as “Improving the genetic traits influencing rooster semen quality and quantity is crucial…….

Line 15: Remove the word “timely”

Line 17: Replace “This approach will ultimately lead to healthier” with “This approach leads to healthier, more productive….”

Abstract:

Lines 22-23: Rewrite the aim of the study as “The aim of this study was to develop a genetic model to estimate parameters guiding the selection of individual Thai native roosters”

Lines 23-24: Rephrase the sentence to “Using data collected from 3475 records of Thai native roosters aged 1–4 years, we evaluated semen traits (mass movement, semen volume, and sperm concentration) over 54 weeks.” for clarity

Line 26: Replace the word “such as” with “including”

Line 31: Write this way ……...respectively, “indicating moderate genetic influence on these traits.”

Introduction:

Line 47: Replace the word “exhibit with “has”

Lines 52-53: Rephrase as “These native breeds are an important genetic resource for global food security due to their adaptability and diversity.”

Lines 55-56: Rephrase as "However, reproductive traits are often overlooked, resulting in reduced fertility and lower chick production."

Lines 62-65: The sentence is too lengthy to understand. Please split it into two separate sentences.

Lines 69-72: Similarly, this sentence needs to be shorter and easier to understand. It should be split into two separate sentences.

Line 75-77: Add the reference

Materials and methods:

Line 129: Why samples were collected for 54 weeks?

Although the authors have provided information regarding humidity and temperature, how were these parameters measured and controlled?

How were the models used in this study validated?

Results:

Although the statistical analysis provided valuable information regarding semen traits, the R² values for some characteristics, such as semen volume (0.3860) and sperm concentration (0.3732), are very low. This indicates a limitation in the precision of the predicted model.

Moreover, while the SP8 model demonstrated good predictive ability, the heritability estimates remain modest, indicating its limited influence on semen traits. How will you address these points? Additionally, there may be environmental impacts that could be explored.

Discussion:

Conclusion:

Provide a concise conclusion that directly reflects the study's findings. Specify how the models led to unique insights.

Comments on the Quality of English Language

Improve the grammar and sentence structure to make it easier for readers to understand.

Author Response

To Reviewer,

We are grateful for your critical reading and efforts to improve the quality of the manuscript. We hope that the revised manuscript will meet your expectations. Our responses to each comment are listed below.

General Comments:

The article "Unveiling the Genetic Architecture of Semen Traits in Thai Native Chickens: A Comprehensive Analysis Using Random Regression and Spline Function Models" provides valuable insights into the semen traits of Thai native chickens using various statistical models. However, the following points need to be addressed:

Response 1:  We are grateful for your critical reading and efforts to improve the quality of the manuscript. We hope that the revised manuscript will meet your expectations. Our responses to each comment are listed below.

How can the findings of this study be incorporated into breeding programs?

Response 2:  A random regression model with SP8 function from this study allows for more accurate estimation of heritability for semen traits such as mass movement, semen volume, and sperm concentration in Thai native roosters. This enables breeders to identify roosters with superior reproductive traits, improving breeding selections. Additionally, using selection indexes to consider genetic improvement in all three semen traits simultaneously is another viable approach for genetic selection.   

What are the limitations of the SP8 model compared to other models?

Response 3:  As your suggestion, we added the following sentences to explain the limitations of the SP8 model compared to other models: “When using the SP8 model, several factors must be considered. The linear spline with eight knots is more complex than simpler models like SP4 or SP5, which can make it more computationally demanding and harder to implement with smaller datasets or limited computing resources. Additionally, the increased number of knots may lead to overfitting, particularly with datasets that have fewer observations, potentially limiting the model's generalizability to other populations or time periods. Therefore, it is essential to validate the model's goodness-of-fit to ensure its suitability for estimating genetic parameters in animal breeding.” See lines 479-486.

Are the findings of this study applicable to different poultry species and breeding environments?

Response 4:  The use of spline functions and random regression models for estimating genetic parameters in Thai native roosters can also be applied to other poultry species. Key genetic principles, such as heritability and genetic correlations, are broadly relevant across species. This study has important implications for both local and global breeding programs. By improving the accuracy of genetic parameter estimates, breeders can make more informed decisions to enhance reproductive traits, such as semen quality and quantity. These advancements not only benefit Thai native chickens but also offer insights that can improve reproductive efficiency in other poultry species, leading to better productivity, sustainable farming, and economic gains in diverse breeding systems.

Simple summary:

Lines 11-12: It should be written as “Improving the genetic traits influencing rooster semen quality and quantity is crucial…….

Response 5: We replaced the sentence as your suggestion “Improving the genetic traits influencing rooster semen quality and quantity is crucial……. See lines 11-12.

Line 15: Remove the word “timely”

Response 6: We removed the word “timely” from the revised manuscript. See line 15.

Line 17: Replace “This approach will ultimately lead to healthier” with “This approach leads to healthier, more productive….”

Response 7: We replaced the sentence as your suggestion. See line 17.

Abstract:

Lines 22-23: Rewrite the aim of the study as “The aim of this study was to develop a genetic model to estimate parameters guiding the selection of individual Thai native roosters”

Response 8: done as requested. See lines 22-23.

Lines 23-24: Rephrase the sentence to “Using data collected from 3475 records of Thai native roosters aged 1–4 years, we evaluated semen traits (mass movement, semen volume, and sperm concentration) over 54 weeks.” for clarity

Response 9: done as requested. See lines 23-25.

Line 26: Replace the word “such as” with “including”

Response 10: done as requested. See line 26.

Line 31: Write this way ……...respectively, “indicating moderate genetic influence on these traits.”

Response 11: done as requested. See lines 31-32.

Introduction:

Line 47: Replace the word “exhibit with “has”

Response 12:  done as requested. See line 46.

Lines 52-53: Rephrase as “These native breeds are an important genetic resource for global food security due to their adaptability and diversity.”

Response 13: done as requested. See lines 51-52.

Lines 55-56: Rephrase as "However, reproductive traits are often overlooked, resulting in reduced fertility and lower chick production."

Response 14:  done as requested. See lines 54-55.

Lines 62-65: The sentence is too lengthy to understand. Please split it into two separate sentences.

Response 15:  done as requested. See lines 60-64.

Lines 69-72: Similarly, this sentence needs to be shorter and easier to understand. It should be split into two separate sentences.

Response 16: done as requested. See lines 68-71.

Line 75-77: Add the reference

Response 17: References have been added to the revised manuscript as detailed below. See line 76 and reference section.  

  1. Parker, H.M.; McDaniel, C.D. Selection of young broiler breeders for semen quality improves hatchability in an industry field trial. Appl. Poult. Res. 2002, 11, 250–259.
  2. Liang, W.; He, Y.; Zhu, T.; Zhang, B.; Liu, S.; Guo, H.; Liu, P.; Liu, H.; Li, D.; Kang, X.; Li, W.; Sun, G.Dietary restriction promote sperm remodeling in aged roosters based on transcriptome analysis. BMC Genomics. 2024, 25, 680.

Materials and methods:

Line 129: Why samples were collected for 54 weeks?

Response 18: A 54-week period encompasses a significant portion of the rooster's reproductive life, allowing breeders to assess semen quality and quantity over time and identify any patterns or fluctuations related to age, health, or environmental factors. Additionally, regular collection over this period enables breeders to monitor genetic progress and evaluate the consistency of semen production. This helps in selecting roosters with superior reproductive traits for genetic improvement in the flock. In general, semen quality typically declines with age, especially after the rooster's peak fertility. Prolonged collection over 54 weeks may result in lower-quality semen, potentially reducing fertility rates in artificial insemination programs and affecting the accuracy of genetic parameter estimations.

Although the authors have provided information regarding humidity and temperature, how were these parameters measured and controlled?

Response 19: Ambient temperature and relative humidity in the chicken housing were recorded hourly throughout the experiment using automatic environmental data loggers. This equipment allowed real-time monitoring and historical data analysis to assess the impact of environmental conditions on semen quality and quantity. If any abnormalities in the semen were observed, the recorded environmental data were analyzed to identify potential causes.

How were the models used in this study validated?

Response 20: To ensure the model used for estimating parameters is accurate and appropriate for selecting and improving the genetics of semen quantity and quality in native chickens, we follow a 3-step process (see lines 209-224) as detailed below:

  1. Goodness of fit criteria using mean square error (MSE), the coefficient of determination (R2), twice the negative log likelihood (-2logL), and Akaike’s information criterion (AIC) from the whole data set, in which a lowest MSE, -2logL, and AIC and highest R2 indicate the best-fit model. The MSE and R2 are defined as: MSE = , where is the error sum of the square,  is the number of observations,  is the number of model parameters, R2 = , where  is the total sum of squares, -2logL = , where  denotes the vector of the model parameters, is the likelihood of the data y evaluated at the maximum likelihood estimate , AIC = , where  is the number of model parameters.
  2. The predictive ability in terms of through cross-validation, in which higher values indicate better predictive ability.
  3. The heritability value (h2), when using any genetic model that gives a high heritability value, means that most of the variation in the trait is due to genetic factors. This makes genetic selection more efficient because the traits selected are more likely to be passed on to the next generation and can also lead to a reduction in the generation interval (the average age of parents when their offspring are born) because it allows for early selection.

Results:

Although the statistical analysis provided valuable information regarding semen traits, the R² values for some characteristics, such as semen volume (0.3860) and sperm concentration (0.3732), are very low. This indicates a limitation in the precision of the predicted model.

Response 21: While low R² values typically indicate limited accuracy in model predictions, the R² values in this study still provide valuable insights, particularly in identifying the best-fitting genetic model. The highest R² values (observed in SP8 for all traits studied) suggest that the model fits the data well and accurately predicts the traits by accounting for genetic and environmental factors. On the other hand, the lower R² values, such as those for semen volume and sperm concentration, indicate the presence of additional factors, likely environmental, that need to be considered beyond genetics.

In genetic evaluations, R² assesses the reliability of estimated breeding values (EBVs). Low R² values indicate limitations in the model's predictive power, suggesting the need for more data, additional variables, or alternative modeling approaches to improve accuracy. Understanding R² allows breeders to gauge the effectiveness of their genetic models in selecting desirable traits.

However, this study did not rely solely on the R² statistic. Other metrics, including mean square error (MSE), twice the negative log likelihood (-2logL), and Akaike’s information criterion (AIC), were also used to ensure accurate identification of the best-fit genetic model and precise estimation of genetic parameters.

Moreover, while the SP8 model demonstrated good predictive ability, the heritability estimates remain modest, indicating its limited influence on semen traits. How will you address these points? Additionally, there may be environmental impacts that could be explored.

Response 22: Based on your suggestions, we have written the following additional sentence: “Although the genetic models used in this study indicated that the estimated genetic parameters were still low, integrating other approaches, such as marker-assisted selection (MAS), quantitative trait loci (QTL), or genomic selection (GS), along with improving nutrition, health management, and environmental factors, may enhance the expression of genetic potential. By combining these methods, the effectiveness of breeding programs can be improved even when genetic parameters are low, ensuring continued progress in selecting superior animal breeds.” See lines 487-493.

Discussion:

Conclusion:

Provide a concise conclusion that directly reflects the study's findings. Specify how the models led to unique insights.

Response 23: We have rewritten the conclusion section to be more concise and clearer as follows: “The results of this study confirm that spline functions provide the best fit for estimating the genetic parameters of semen traits in Thai native roosters. A random regression test-day model with eight knots linear spline function best described the heritability curve over the semen collection period. Therefore, it is possible to obtain estimated breeding values to improve both the quality and quantity of semen in breeding programs for Thai native chickens. These findings will improve selection strategies for reproductive performance, ultimately benefiting the poultry industry and promoting sustainable practices. Moreover, the implications of this research extend beyond local applications, providing valuable insights for genetic studies in various poultry species worldwide.” See lines 498-506.

Comments on the Quality of English Language

Improve the grammar and sentence structure to make it easier for readers to understand.

Response 24: We have already checked and edited the English grammar carefully, as you suggested.

Best Regards

Wuttigrai Boonkum

Reviewer 2 Report

Comments and Suggestions for Authors

Studying the genetic characteristics of the quality and quantity of rooster sperm is one of the most important tasks for optimizing poultry production. In this study covers a large period, the authors have done a lot of work on monitoring the parameters of semen quality in roosters of thai native chickens. The dependence of the influence of the rooster's age on the parameters of semen quality is shown. The content of the article can help change the approach to evaluating semen not only of thai native chickens but also of other breeds. Overall, the article leaves a positive impression. The conducted study will allow the maximum long-term use of roosters to obtain high-quality semen. 

A topical question was being developed on the study of genetic traits of chicken semen. This article presents data that helps to improve and refine poultry breeding programs. This article is a scientific study and similar studies have not been conducted before. This article presents an innovative approach to the study of genetic parameters of chicken semen, which allows changing the approach to chicken evaluation. The authors used modern methodological approaches and statistical methods of analysis, which allow obtaining high-quality results.   The presented findings are fully consistent with the goals and objectives of the study and the support for the question posed. The study uses modern methodological approaches that allow achieving the goal set by the authors, in addition, the presented review literature sufficiently fully describes the problem posed to the authors. The list of references used in this article is fully appropriate in the amount of 69 sources The graphs and tables presented in the manual fully reflect the results obtained.

A wish to the authors: in the continuation of the work, add an analysis of genomic data.

But there are a number of questions:

It is not entirely clear from the materials and method how many roosters participated in the study. Records of 3475 birds are mentioned, from all of them or from some of them semen was obtained.

How representative is the sample of roosters used in the study?

Author Response

To Reviewer,

We are grateful for your critical reading and efforts to improve the quality of the manuscript. We hope that the revised manuscript will meet your expectations. Our responses to each comment are listed below.

Response to Reviewer 2 Comments

Studying the genetic characteristics of the quality and quantity of rooster sperm is one of the most important tasks for optimizing poultry production. In this study covers a large period, the authors have done a lot of work on monitoring the parameters of semen quality in roosters of Thai native chickens. The dependence of the influence of the rooster's age on the parameters of semen quality is shown. The content of the article can help change the approach to evaluating semen not only of Thai native chickens but also of other breeds. Overall, the article leaves a positive impression. The conducted study will allow the maximum long-term use of roosters to obtain high-quality semen. 

A topical question was being developed on the study of genetic traits of chicken semen. This article presents data that helps to improve and refine poultry breeding programs. This article is a scientific study and similar studies have not been conducted before. This article presents an innovative approach to the study of genetic parameters of chicken semen, which allows changing the approach to chicken evaluation. The authors used modern methodological approaches and statistical methods of analysis, which allow obtaining high-quality results.   The presented findings are fully consistent with the goals and objectives of the study and the support for the question posed. The study uses modern methodological approaches that allow achieving the goal set by the authors, in addition, the presented review literature sufficiently fully describes the problem posed to the authors. The list of references used in this article is fully appropriate in the amount of 69 sources The graphs and tables presented in the manual fully reflect the results obtained.

A wish to the authors: in the continuation of the work, add an analysis of genomic data.

Response 1:  

We are grateful for your critical reading and efforts to improve the quality of the manuscript. We hope that the revised manuscript will meet your expectations. Our responses to each comment are listed below.

But there are a number of questions:

It is not entirely clear from the materials and method how many roosters participated in the study. Records of 3475 birds are mentioned, from all of them or from some of them semen was obtained.

Response 2: We have revised the wording to make it clearer as follows “The data consisted of 3475 records of 242 Thai native grandparent roosters (Pradu Hang Dum)…”. See lines 118-119. 

How representative is the sample of roosters used in the study?

Response 3:  In this study, we used all 242 grandparent roosters available on our farm. This number is sufficient to represent the grandparent population for producing the next generation of chickens. According to theory, at least 100 grandparent roosters should be maintained on the farm for selection and breeding purposes.

Best Regards

Wuttigrai Boonkum

Round 2

Reviewer 1 Report

Comments and Suggestions for Authors

Thank you for revising the manuscript and addressing all the points raised in the initial review. I am satisfied with the revised version.